# Determining the Role of Acidity, Fate and Formation of IEPOX-Derived SOA in CMAQ

**Petros Vasilakos [1,2,*], Yongtao Hu [2], Armistead Russell [2] and Athanasios Nenes [3,4,*]**

1   School of Chemical and Biomolecular Engineering, Georgia Institute of Technology, Atlanta, GA 30332, USA
2   School of Civil and Environmental Engineering, Georgia Institute of Technology, Atlanta, GA 30332, USA;
    yh29@mail.gatech.edu (Y.H.); ar70@gatech.edu (A.R.)
3   Institute of Chemical Engineering Sciences, Foundation for Research and Technology-Hellas,
    GR 26504 Patras, Greece
4   School of Architecture, Civil & Environmental Engineering, École Polytechnique Fédérale de Lausanne,
    1015 Lausanne, Switzerland
*   Correspondence: peter.vasilakos@gatech.edu (P.V.); athanasios.nenes@epfl.ch (A.N.)

**Abstract:** Formation of aerosol from biogenic hydrocarbons relies heavily on anthropogenic emissions since they control the availability of species such as sulfate and nitrate, and through them, aerosol acidity (pH). To elucidate the role that acidity and emissions play in regulating Secondary Organic Aerosol (SOA), we utilize the 2013 Southern Oxidant and Aerosol Study (SOAS) dataset to enhance the extensive mechanism of isoprene epoxydiol (IEPOX)-mediated SOA formation implemented in the Community Multiscale Air Quality (CMAQ) model (Pye et al., 2013), which was then used to investigate the impact of potential future emission controls on IEPOX OA. We found that the Henry's law coefficient for IEPOX was the most impactful parameter that controls aqueous isoprene OA products, and a value of $1.9 \times 10^7$ M atm$^{-1}$ provides the best agreement with measurements. Non-volatile cations (NVCs) were found in higher-than-expected quantities in CMAQ and exerted a significant influence on IEPOX OA by reducing its production by as much as 30% when present. Consistent with previous literature, a strong correlation of isoprene OA with sulfate, and little correlation with acidity or liquid water content, was found. Future reductions in SO$_2$ emissions are found to not affect this correlation and generally act to increase the sensitivity of IEPOX OA to sulfate, even in extreme cases.

**Keywords:** IEPOX; SOA; Henry's Law; aerosol pH; acidity; emissions reductions; non-volatile cations; sensitivity



## 1. Introduction

Isoprene (C$_5$H$_8$) is the most common abundant biogenic volatile organic compound (BVOC) emitted by foliage and trees [1], constituting a significant contributor to the total SOA load especially in the Southeastern (SE) US, where observations have found that isoprene-derived OA can comprise up to 30% of the total organic aerosol (OA) [2]. Anthropogenic pollutants interact with isoprene gas phase products [2,3] through a multi-step chemical mechanism, as seen from correlations between biogenic SOA (BSOA) and sulfate which support this finding [2–5].

One of the main gas-phase oxidation products of isoprene are isoprene epoxydiols (IEPOX), which are a fourth-generation product of isoprene's reaction with the hydroxyl radical. The proposed mechanism requires that IEPOX diffuses from the gaseous phase in an acidic, aqueous environment, where it reacts with a nucleophile such as SO$_4^{-2}$ or OH$^-$ to produce SOA [6]. This SOA exhibits an almost linear association with the amount of available sulfate [2], while, at the same time, showing little dependence on liquid water content and acidity, suggesting that sulfate is the controlling factor for the amount of isoprene OA being produced in the SE US. Given that natural emissions of BVOCs such

as isoprene and terpenes in the SE are the highest in the continental US and coupled with significant anthropogenic emission sources in the vicinity such as power plants and automobiles, increased aerosol formation is expected [7]. The interactions between the anthropogenic and biogenic species have been extensively studied in recent years [2,4,5,8]; the underlying mechanisms of SOA formation however, are still not fully constrained since parameters controlling specific reaction steps remain uncertain [9,10].

Biogenically-derived SOA in the Southeast US can comprise up to half of the total organic mass in the summertime and up to 30% year-around [3,7,11,12]. Terpene species such as $\alpha$-pinene and $\beta$-pinene, are significant contributors to the mass of SOA, since they react very fast with atmospheric oxidants with high yields [13]. In addition, recent studies [14] have shown that isoprene is also producing SOA with an appreciable mass yield, mostly through the IEPOX pathway, with the major products being 2-methyltetrols, organosulfates and to a lesser extent, organonitrates [4,5,8]. Given the high levels of aerosol water, low aerosol pH and high $SO_2$ emissions [15,16], high levels of IEPOX OA are expected, something that is confirmed by measurements [2] and modelling studies [8].

In the Southeastern US, variability in IEPOX OA formation is not correlated with inorganic aerosol acidity [2,12]. However, this is the case when water and sulfate availability are high and, therefore, future reductions in $SO_2$ emissions which will inevitably reduce the total amount of available sulfate and liquid water can lead to different behavior, potential forcing IEPOX OA formation to a regime where acidity plays a larger role. Aerosol pH is largely insensitive to $SO_2$ reductions [16,17] due to the buffering effect of semi-volatile ammonia that repartitions between the gas/particle phase in response to changes in sulfate—so that very large reductions are required in order for pH to appreciably elevate. Even a unit increase, however, can lead to a completely different aerosol composition, given the impact of acidity on nitrate partitioning and IEPOX OA formation [8,18]. From a modelling perspective, the presence of non-volatile cations (NVCs) can also introduce biases in pH predictions, further increasing the uncertainty in IEPOX OA predictions [18].

In this study, we utilize the observations from the Southern Oxidant and Aerosol Study (SOAS) campaign taken at the Centerville, Alabama, Southeastern Aerosol Research and Characterization (SEARCH) [19,20] monitoring location, which took place between 1 June and 15 July 2013. The objectives of the campaign are detailed in Carlton et al. (2013), with a strong focus on investigating the interactions between anthropogenic emissions and BVOCs. Since the campaign took place in a densely forested area, the emissions of BVOCs such as isoprene, $\alpha$-pinene and $\beta$-pinene, were expected to be high and therefore play an important role in the region's climate. High concentrations of SOA were observed, and SOA resulting from isoprene oxidation alone exhibited concentrations of the order of 2 $\mu$g m$^{-3}$ [2,4].

With the utilization of existing measurements of gas-phase precursors and aerosol, we evaluate and improve the isoprene OA formation mechanism in the Community Multiscale Air Quality (CMAQ) model, by constraining reaction rate constants for isoprene oxidation, as well as partitioning parameters such as the uncertain Henry's law constant. CMAQ exhibits a negative bias in modelled OA concentrations [8], especially SOA from isoprene oxidation. The IEPOX pathway has been implemented in CMAQ [8] and the Goddard Earth Observing System Chemistry Model (GEOS-Chem) [21], but large uncertainties remain in system controlling parameters. This work aims at performing an extensive comparison of model results to data, and calibration of model constants to enable the model to reproduce the observed correlation between sulfate and isoprene SOA, as well as reduce the bias between predicted and observed organic matter. Given the close relationship between sulfate and IEPOX OA [2], and the expectation that $SO_2$ emissions will keep declining in the future, the updated model is then used to study the response of IEPOX OA to further emission reductions, as well as quantify the resulting pH changes due to the aforementioned reductions. Specifically, the changes incurred by $SO_2$ reductions on pH, liquid water content (LWC) and IEPOX OA were studied to determine if acidity will indeed play a role in regulating the future of IEPOX OA in the SE US.

## 2. Materials and Methods

### 2.1. Chemical Transport Model and Measurements

For the needs of this work, the CMAQ model was used [22]. CMAQ is a three-dimensional, Eulerian, atmospheric chemistry and transport model, that simulates the processes atmospherically relevant compounds undergo such as emission, diffusion, chemical reactions and deposition. A version of CMAQ with extended isoprene chemistry (Pye et al. 2013—version 5.0.2) was employed, that included formation of IEPOX OA (hereafter IEPOX OA) from the reactive uptake of isoprene epoxides into aqueous phase aerosol. The aqueous phase formation of IEPOX OA, which for the needs of this study was defined as the sum of organonitrates (ON), organosulfates (OS) and methyltetrols (MT), is controlled by the IEPOX uptake coefficient [8], which is a function of Henry's law coefficient (H). Accurate representation of H is important, since it controls aqueous IEPOX availability and can control the phase separation of multiphase aerosols. Aerosol pH was calculated inline using ISORROPIA v2.2 [23].

Simulations were conducted using a 36 km × 36 km resolution grid over the continental US for the SOAS campaign period, with 5 days used as a start-up. Meteorological data were developed using the Weather Research Forecasting (WRF) model and converted to CMAQ inputs using the Meteorology-Chemistry Interface Processor (MCIP). Unless otherwise stated, all results shown in this paper are specific to the grid cell that includes the Centreville site, to allow for comparison with the SOAS measurements.

The base biogenic emissions were calculated online using the Biogenic Emission Inventory System (BEIS) [24]. BEIS uses the meteorological data in order to obtain variables governing the magnitude of BVOC emissions, such as solar irradiance (SR), temperature (T), as well as land data (plant type and plant coverage of each grid cell) to determine the fluxes of BVOCs. Non-volatile cations have a significant impact on pH and uncertainties in their emissions and size dependence can lead to biases in CMAQ [18], albeit with the largest overestimates occurring during the night when IEPOX OA chemistry is much less active than in daytime. For this reason, they were removed from the simulations to avoid biasing the results.

Multiple simulations were conducted in order to gauge the ability of the model to reproduce the measurements [6,14,25,26] and the correlations that govern them, and parameters and inputs associated with IEPOX OA formed via the IEPOX mechanism were modified accordingly.

After updating the model to a point where it was capable of capturing the observed behavior of IEPOX OA, sensitivity tests were conducted. Specifically, the role of NVCs on pH and subsequently IEPOX OA was quantified by comparing predicted aerosol in their presence and absence. To simulate future emission reductions scenarios, total $SO_2$ emissions across the US were reduced by 25%, 50%, 75% and 100% and their impact on IEPOX OA was quantified.

### 2.2. Chemical Mechanism

Chamber experiments and field observations find that the dominant pathway for the formation of SOA through isoprene involves the formation of intermediate IEPOX, [2,4,5,9,10]. In recent modelling studies [8,21,27], a similar mechanism was introduced in CMAQ and GEOS-Chem, using IEPOX as an intermediate for isoprene OA formation and found that 15–20% of the total SOA can be attributed to isoprene oxidation products, with the newer 5.2 and 5.3 CMAQ versions offering better performance [27,28]. The mechanism as is implemented in the CMAQv5.0.2 is described in Figure 1.

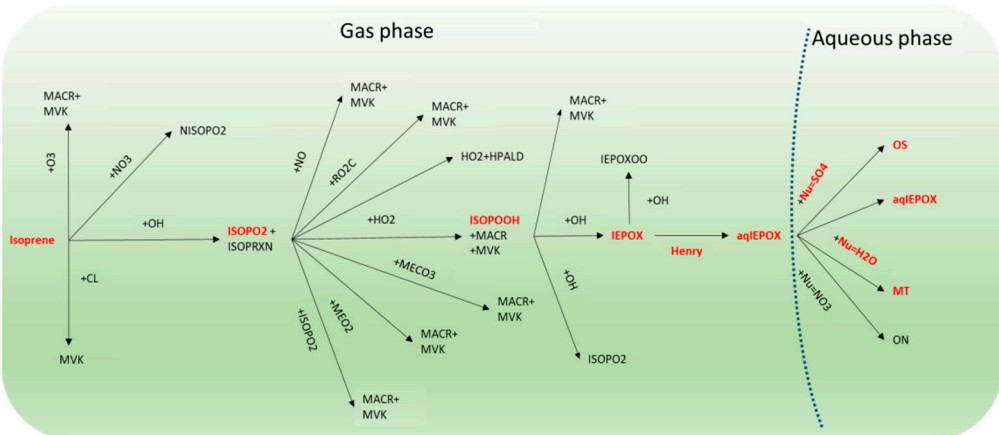

**Figure 1.** Isoprene epoxydiol organic aerosol (IEPOX OA) formation mechanism as implemented in the latest version of the Community Multiscale Air Quality (CMAQ) model (Pye et al. 2013). OS, MT and ON stand for organosulphates, methyltetrols and organonitrates respectively.

## 3. Results

### 3.1. Ozone, NOx and Sulfate

In all simulations including the base scenario, there was good agreement between the measurements and the simulated values of atmospherically relevant compounds at the Centreville site. Ozone was strongly correlated with the measurements, but exhibits a consistent positive bias of approximately 10 ppb (Figure 2a), while $NO_x$ was captured well, albeit with less variability than the measurements (Figure 2b). Reasons for positive biases in model simulations of ozone in the SE US have been explored in Travis et al. 2016 [29] and could include but are not limited to errors in vertical mixing and production rates within the planetary boundary layer height (PBL). For aerosol species, sulfate closely tracked the measurements (Figure 2d), but was 26% lower during the middle of the day. Although there was an appreciable amount of isoprene OA predicted for the SE US, for the SOAS site IEPOX OA was much lower than the amount estimated from the Positive Matrix Factorization (PMF) factors derived from the measurements by almost 1.2 μg m$^{-3}$. All of the above species, except for IEPOX OA, changed little with the modifications discussed below, while for the optimal H simulation it was about 30% lower than the PMF factor for SOAS [2] (Figure 2c). The initial simulation, where only semivolatile isoprene OA could form (default version of CMAQ without IEPOX extended chemistry), produced very little OA (Figure 2c—green line), indicating that semivolatile OA was only a small fraction of total isoprene OA.

### 3.2. Henry's Law Sensitivity Tests and Updates to the Simulations

#### 3.2.1. Baseline Simulation

Our initial simulation (hereafter baseline) used the default version of CMAQ as described in Pye et al. 2013. For this case, the Henry's law coefficient for IEPOX was set to $2.7 \times 10^6$ M atm$^{-1}$. Biogenic emissions for isoprene were not changed and left to the value generated by BEIS. The deposition surrogate used to calculate the dry deposition of IEPOX was methylhydroxyperoxide (CMAQ species VD_OP), with a relatively low H of $3.1 \times 10^2$ M atm$^{-1}$. The PBL was calculated by the WRF meteorology and used as is.

Results from the baseline simulation showed that IEPOX OA was severely underestimated, especially methyltetrols (MT) and organosulfates (OS). Isoprene levels were biased low during the day time and exhibited a night time high, suggesting that the isoprene emissions from BEIS were not accurate (Figure 3). However, IEPOX levels were overestimated when compared to the observations by a factor of 10. The relative ratios of IEPOX-derived OA (OS to MT) compared favorably with the observations, suggesting that the aqueous chemical mechanism represents the underlying physics accurately. Similarly, important gas phase and aerosol species, such as ozone, $NO_x$ and sulfate were accurately predicted on

average, although with some bias, possibly due to the long simulation period. To alleviate the issues identified with the simulation, we applied a number of sequential updates to the model by making use of the available measurements, in order to ensure that the gas phase products were as close as possible to the observed values before attempting any changes in the aqueous chemistry.

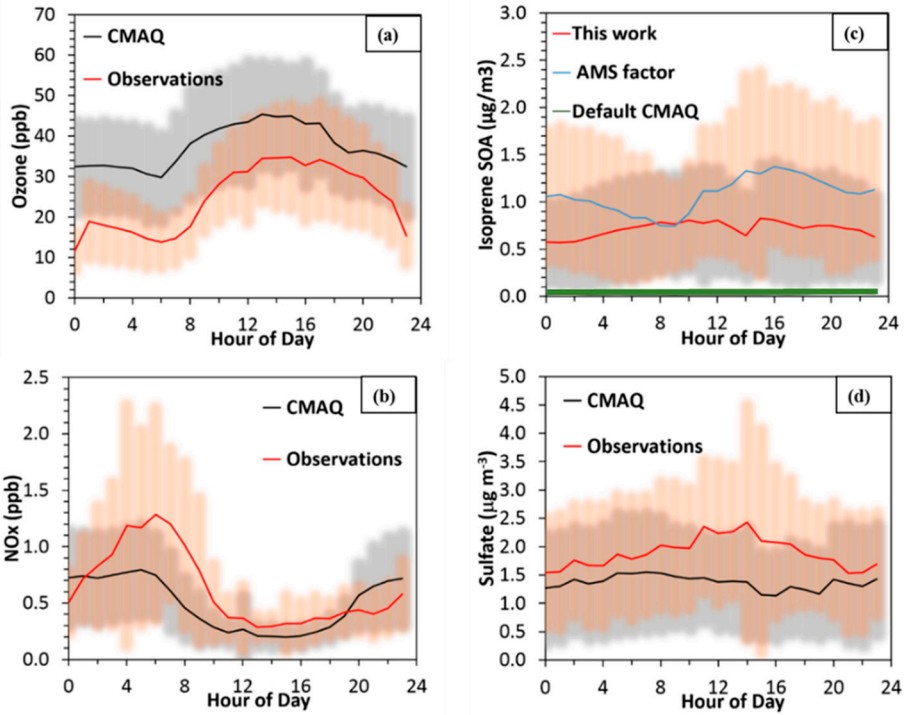

**Figure 2.** Diurnal profiles during the Southern Oxidant and Aerosol Study (SOAS) campaign for measured (red), simulated (black) and default CMAQ (green) for (**a**) ozone, (**b**) NO$_x$, (**c**) isoprene SOA and (**d**) sulfate. The shaded areas represent one standard deviation at each diurnal hour. The default CMAQ 5.0.2, which includes only semivolatile isoprene OA (green) leads to very little isoprene SOA in (**c**).

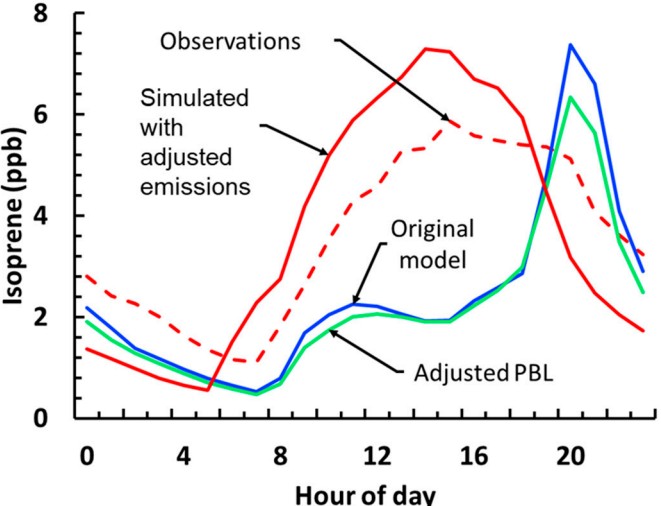

**Figure 3.** Observed (dashed-red), default model (blue), adjusted planetary boundary layer height (PBL, green) and adjusted PBL and emissions (solid red) isoprene diurnal profiles for the Centreville gridcell. Concentrations at each hour correspond to the campaign average for that hour.

### 3.2.2. Planetary Boundary Layer Height (PBL) Prescription and Isoprene Emission Fitting (IEF)

In order to rectify the unrealistic isoprene profile, the first change we applied to the model was direct the assimilation of PBL data available from SOAS, by replacing the model-predicted values with the measured ones. The PBL height predicted from WRF was biased high during the day and biased low during the night compared to observations, which is a known issue with WRF [30] that could explain the isoprene daytime low and its nighttime high. Results from the PBL simulation indicated that this change had little impact on all important tracers. Daytime isoprene levels remained similar to the baseline simulation, while there was a slight decrease of the nighttime high by 1 ppb (Figure 3). The PBL simulation indicated that the behavior of the isoprene levels was driven by the emissions and not by the meteorology, corroborated by simulated temperature profiles that closely matched the observed profiles.

Isoprene emissions in our simulations were under-predicted during the day time and overestimated during the night time, which led to the spike in isoprene concentration during the night (Figure 3). To resolve the issue, the measured isoprene emissions as well as the emissions predicted by BEIS were used, in order to determine scaling factors with which modelled isoprene emissions were multiplied at each model timestep as to better match the observed and simulated levels. As an initial guess the ratio of the measured to simulated values was used, which was then optimized through multiple linear regressions, to achieve better agreement between model and measurements. This simulation (referred to as isoprene emissions flux, or IEF) significantly improved isoprene levels and eliminated the daytime low and the nighttime high (Figure 3).

### 3.2.3. Isoprene Epoxydiol (IEPOX) Deposition Correction (DEP)

While the IEF simulation improved isoprene levels, the extreme overestimation of IEPOX still remained. Dry deposition for IEPOX is expected to be a significant loss process, given the stickiness of the molecule and propensity to deposit to wet surfaces [31]; however, the deposition surrogate used in CMAQ was methylhydroxyperoxide, a slowly depositing molecule, which has a Henry's law constant four orders of magnitude less than that of IEPOX, meaning that depositional loss of IEPOX to wet surfaces was most likely significantly underestimated (depositional time scale of $\tau = 11$ h). By changing the surrogate to $HNO_3$ in the deposition-adjusted (DEP) update, the timescale was reduced by 50% and there was a marked decrease of IEPOX levels to half of what they originally were. Surprisingly, IEPOX OA levels were not impacted by this change, suggesting that other processes were rate-controlling with regards to SOA production.

### 3.2.4. Updated IEPOX Gas Phase Oxidation and Henry's Law Sensitivity Tests

After both the IEF and DEP changes, the IEPOX levels were still positively biased when compared to observations. A potential reason could be the underestimation of the gas phase IEPOX oxidation loss to OH. The rate constant used for the gas phase loss of IEPOX for the previous simulations was $1.5 \times 10^{-1}$ cm$^3$ molecules$^{-1}$ s$^{-1}$ [31]. We updated the rate constant for the gas phase loss of IEPOX to OH, to a value of $3.6 \times 10^{-11}$ cm$^3$ molecules$^{-1}$ s$^{-1}$ [32]. The updated oxidation simulation, slightly reduced IEPOX levels by 5%, while keeping IEPOX OA levels approximately constant. The IEPOX overestimation remained (Figure 4) which implies that the existing sinks are still not significant enough or there is another removal process (for example an additional gas phase reaction) which is not included in the model.

After the above changes, and ensuring that there is sufficient gas phase IEPOX available to react in the aqueous phase, the negative bias for IEPOX OA persisted and the linear relationship between sulfate and IEPOX OA was still not captured [2]. The IEPOX Henry's law constant is one of the most uncertain parameters of the system [33] and at the same time one the most important ones, since it directly controls the amount of IEPOX that is available in the aqueous phase to produce SOA. The literature reported range for the Henry's law constant for IEPOX spans more than 2 orders of magnitude [8,33,34], so a

number of sensitivity tests were performed to estimate a value led to the most consistent results between the model and measurements, using as a constraint the correlation between sulfate and IEPOX OA from Xu et al. (2015), since it is a better indicator for the accuracy of the chemical processes included in the model than the concentration of IEPOX OA. We observed an almost linear relationship between the average levels of isoprene OA and the logarithm of H (Figure 5). A value of $1.9 \times 10^7$ M atm$^{-1}$, which is similar to recent estimates, yields the best overall agreement [4,10,35]. A comprehensive list of simulated scenarios is shown in Table 1.

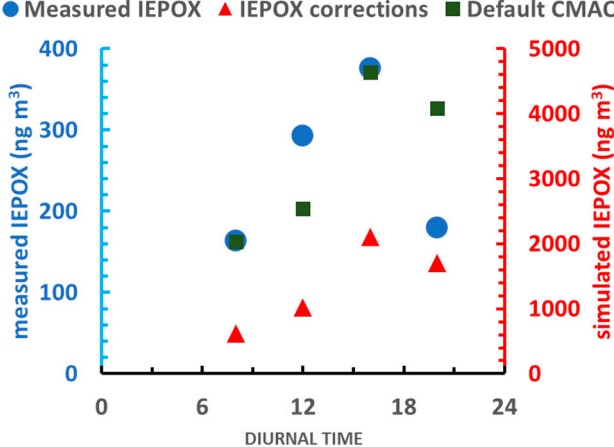

**Figure 4.** Measured (cyan), default CMAQ (green) and corrected (red) IEPOX diurnal concentrations. The IEPOX corrections data refers to IEPOX levels after updating both the deposition surrogate and the reaction rate constant for the OH$^-$ reaction. Measurements from [36].

**Table 1.** List of simulated scenarios and their specifications. The Henry's law coefficient H is equal to $2.7 \times 10^6$ M atm$^{-1}$.

| Simulation Name | PBL Height | Isoprene Emissions | IEPOX Deposition Surrogate | IEPOX + OH$^-$ Oxidation Constant | H* Scaling Factor |
|---|---|---|---|---|---|
| **BASELINE** | default | default | VD_OP | Paulot | 1 |
| **PBL** | assimilated | default | VD_OP | Paulot | 1 |
| **IEF** | assimilated | adjusted | VD_OP | Paulot | 1 |
| **DEP** | assimilated | adjusted | HNO3 | Paulot | 1 |
| **Oxidation** | assimilated | adjusted | HNO3 | Jacobs | 1 |
| **Henry 1** | assimilated | adjusted | HNO3 | Jacobs | 2.5 |
| **Henry 2** | assimilated | adjusted | HNO3 | Jacobs | 5 |
| **Henry 3** | assimilated | adjusted | HNO3 | Jacobs | 7 |
| **Henry 4** | assimilated | adjusted | HNO3 | Jacobs | 9 |
| **Henry 5** | assimilated | adjusted | HNO3 | Jacobs | 10 |
| **Henry 6** | assimilated | adjusted | HNO3 | Jacobs | 100 |

*3.3. Comparing Aqueous SOA to Observations and Correlation with Sulfate*

Recent studies [2,4,5] have observed a strong correlation between sulfate and isoprene OA in the vicinity of the SE US. To test the validity of the isoprene OA production mechanism in the current version of CMAQ, we used the coefficients of this linear relationship (slope and intercept) as the main parameters to be optimized with our Henry's law sensitivity tests. The slope and intercept of the correlation were less susceptible to emission/model biases, since they were process controlled parameters (reaction and diffusion of IEPOX in the aqueous phase) and not governed by concentrations.

In the base case, without any updated or extended isoprene chemistry, there was a correlation between sulfate and isoprene OA; however, the levels of isoprene OA were far

too low and the slope almost 0 (Figure 6). Using an H of $1.9 \times 10^7$ M atm$^{-1}$, the simulated and observed correlations achieve remarkable agreement.

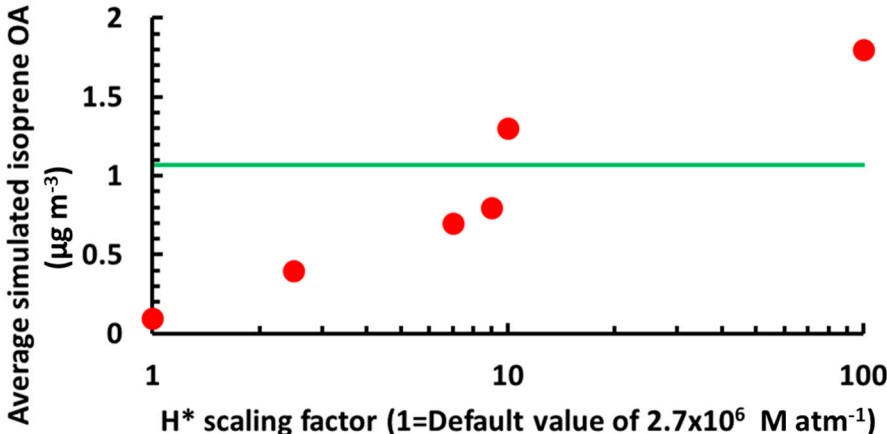

**Figure 5.** Isoprene organic aerosol (OA) sensitivity to H$^+$ from the Henry's law sensitivity tests. The horizontal axis is in units of $2.7 \times 10^6$ M atm$^{-1}$, which corresponds to the default H value in Table 2013. CMAQ version. The green line denotes the average observed isoprene OA.

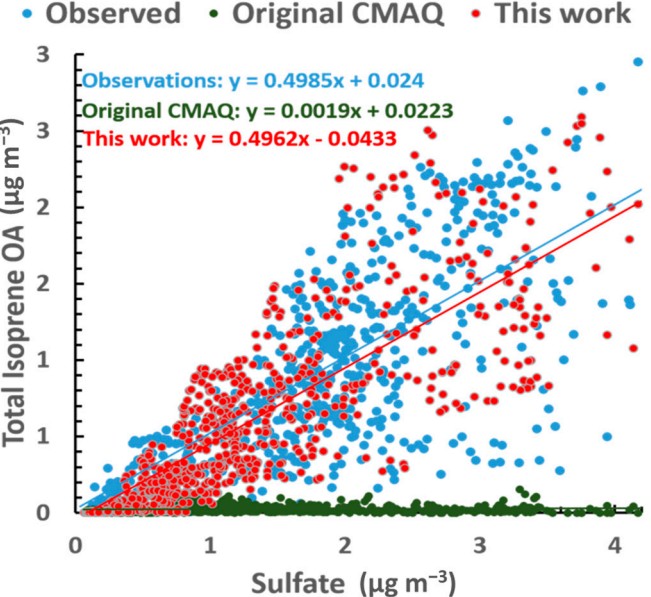

**Figure 6.** Correlation between sulfate and isoprene OA for the SOAS observations (blue), base scenario without updated chemistry (green) and optimal H simulation (red).

The other two important parameters which have been found to control the formation of isoprene OA are acidity (H$^+$) and particle water (H$_2$O$_{ptcl}$) [2,4,6,37]. However, for the case of the SE US, only weak correlations between isoprene OA and H$^+$ or H$_2$O$_{ptcl}$ have been observed [2,4]. In other areas where aerosol pH is higher and water is less abundant, the predicted correlation could be much higher. Note that, for the needs of this analysis and, in the analysis presented in Xu et al. 2015, the definition of acidity used here uses only water as a solvent, which is consistent with recent studies [38] when assuming single-phase chemistry.

There is an abundance of aerosol water in the SE while, at the same time, the mean aerosol pH in Centreville is close to 1 indicating high H$^+$ availability [15,18]. As such, the weak correlation can be explained since, due to their relative abundance, aerosol water and H$^+$ do not constitute limiting parameters for the formation of IEPOX OA, and small changes in their value do not affect isoprene OA levels.

Another possible explanation for the strong correlation with sulfate and the weak correlation with $H^+$ and $H_2O_{ptcl}$, is the competition between acidity and particle water, since increased levels of particle water lead to dilution of ions and reduced pH. In addition, the dilution could weaken a potential salting-in effect, suppressing the IEPOX uptake from the gas phase. While our model does not include this salting-in effect that could be important in some cases, by enhancing the solubility of IEPOX in the aqueous phase, it is not expected to have a strong effect in this study due to the large amounts of water present in the aerosol and the subsequent dilute concentrations of salt ions.

We observe similar behavior in the model when we perform multivariate linear regression on our results, indicating that the current version of the model is able to correctly capture the chemistry behind IEPOX OA production. The regression coefficients are provided in Table 2.

**Table 2.** Results for multiple linear regression of IEPOX OA with respect to sulfate, particle water and $H^+$.

| Regression Variable | Regression Coefficient in the Measurements (Xu et al. 2015) | Regression Coefficient for the Simulations |
|:---:|:---:|:---:|
| **Sulfate** | 0.424 | 0.527 |
| **Water** | −0.004 | 0.029 |
| **$H^+$** | 0.009 | 0.007 |

*3.4. The Impact of Dust on IEPOX Organic Aerosol (OA)*

NVCs are present in all size ranges of atmospheric aerosol, although predominantly found in the coarse mode [18,39,40]. Recently, biases in CMAQ have been identified, where incorrect amounts of NVCs are distributed in the accumulation mode [15,18], leading to aerosol pH predictions that are inconsistent with those obtained using measurement data [15,18]. Given the dependence of IEPOX OA formation on particle acidity, pH biases can also translate to biases in its formation. Because of that, NVCs have been removed from the previous simulations, but a separate simulation to determine their impact was carried out. These are considered bounding simulations i.e., upper and lower limit, since some NVCs are expected to be found in the accumulation mode, albeit in small amounts.

Figure 7 quantifies the impact that NVCs can have on IEPOX OA levels, after all the updates have been implemented on the code, while using the optimal H value. Specifically, when they are excluded from the model (Figure 7a), high concentrations of IEPOX OA are observed in the forested areas of the SE and Eastern US up to 1.8 µg m$^{-3}$ in the Ozarks, Tennessee and Arkansas area, consistent with literature sources [41]. The pH values for these areas are very low ranging from 0 to 1.5, in agreement with the overall trends from Guo et al. 2015. When the CMAQ predicted NVCs are included in the simulation (Figure 7b), the spatial pattern for IEPOX OA remains the same, albeit with marked decreases in concentrations everywhere in the domain. Specifically, the difference between the two simulations is nearly 0.6 µg m$^{-3}$—almost 30% (Figure 7c)—driven by the increased pH in the simulation where NVCs are included, where it's almost 1 to 2 units higher. This suggests that IEPOX OA is not limited by pH under the very acidic conditions in most Southeastern states like Alabama and Georgia, but may be in other states (such as Arkansas) where aerosol is influenced by appreciable amounts of NVCs (Figure 7). Discrepancies in pH predictions is a major source of biases in CMAQ for not only IEPOX OA but also aerosol nitrate [18] and needs to be addressed in a future model update.

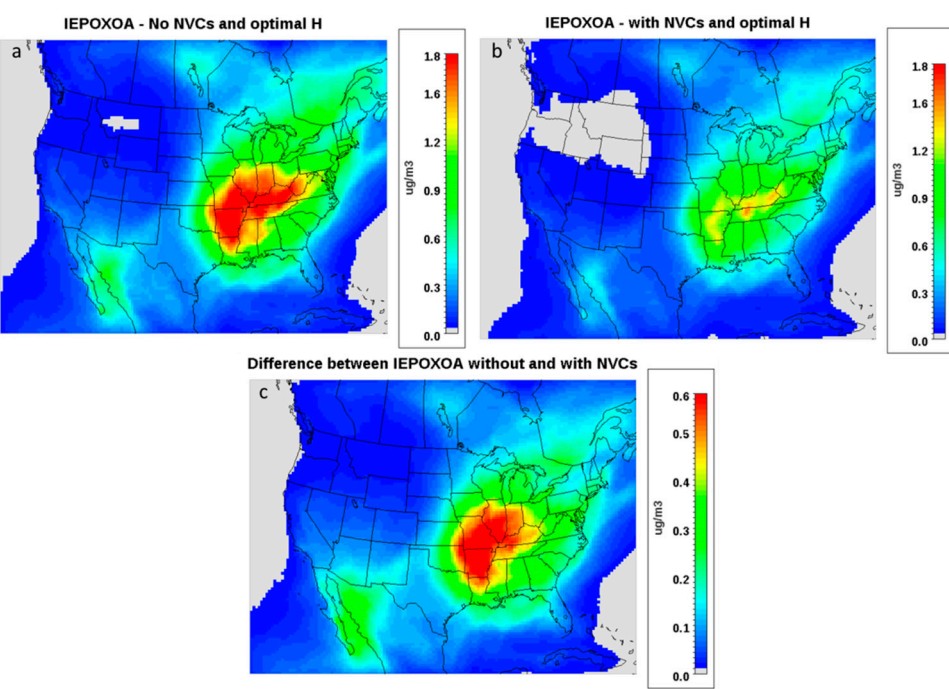

**Figure 7.** Campaign average predicted IEPOX OA when non-volatile cations (NVCs) are excluded (**a**) and included (**b**) in the simulations, as well as the difference (**c**) between the two fields. Simulations for these cases are carried out with all the code updates active.

### 3.5. SO_2 Reductions and the Future of IEPOX OA

Throughout the preceding decade, great strides have been made with regards to improvements on air quality, driven by continuous emission reductions. $SO_2$ in particular has seen decreases of almost 7% per annum [42] for the last decade. Since IEPOX OA formation and $SO_4$ availability are intertwined, it is expected that IEPOX OA levels will drop in response to the $SO_2$ reductions, something that has been corroborated in previous modelling studies looking at the effect of $SO_4$ reductions on IEPOX OA [5,8]. To investigate, we used the updated model and decreased total $SO_2$ emissions by a set amount (25%, 50%, 75% and 100%) to quantify the impact of future reductions on IEPOX OA. These scenarios were simulated for the conditions corresponding to the SOAS campaign (1 June to 15 July 2013). Note that simulations in this work are likely upper bound estimates on the amount of IEPOX formed in the emissions reductions as phase separation and formation of diffusion limited coatings which become more abundant as sulfate is reduced [43] are not considered. LWC drops drastically by 40% in the Eastern US after the first 25% reductions in $SO_2$ emissions (Figure 8a,b). Further emission reductions show diminishing returns in their ability to reduce LWC (Figure 8c–e), and a baseline of about 1 $\mu$g m$^{-3}$ is established under 100% $SO_2$ reductions.

The model predicts acidic pH throughout most of the US in the initial simulation (Figure 9a). Consistent with results from Weber et al. 2016, despite sulfate reductions, the pH in the SE US remains acidic (Figure 9b,c) and only under drastic reductions does the aerosol approach neutrality (Figure 9c,d). It is noteworthy that rural areas of Louisiana, Alabama, Georgia and Florida appear to be less sensitive than the urban centers. This interesting behavior is attributed to the increased nitrate levels in urban areas in these states, when compared to the rural areas, leading to partial substitution of sulfate with nitrate, accelerating the rate by which pH increases in response to $SO_2$ reductions.

For IEPOX OA, the greatest decreases in magnitude are seen for the first 25% in reductions (Figure 10a), with an almost 60% drop in IEPOX OA production over the Eastern and SE US, consistent with results from Budisulistiorini et al. 2017. These decreases are a combination of multiple factors, since the removal of sulfate from the system translates to slightly more alkaline aerosol, coupled with markedly reduced liquid water availability (Figure 8a,b; Figure 9a,b) and less surface area for uptake. Further reductions (Figure 10b–d) also affect IEPOX OA levels, but the drop in IEPOX OA levels is not as pronounced. This is because the removal of sulfate rapidly reduces particle water until a threshold value of about 1 $\mu g\,m^{-3}$ is achieved when no sulfate exists in the system. While the drop in water concentrations is the highest for the first 25% of removed $SO_2$, it decreases in magnitude when more sulfate is removed (Figure 8c–e). When approaching 100% $SO_2$ removal, production of OS is shut down and the only constituents of IEPOX OA is methytetrols and a negligible amount of organonitrates (ON $\leq 0.01\ \mu g\,m^{-3}$). The other isoprene OA pathways from methacrylic acid epoxide (MAE) as well as hydroxymethylmethyl-$\alpha$-lactone (HMML) are still producing approximately the same amount of aerosol throughout all simulations.

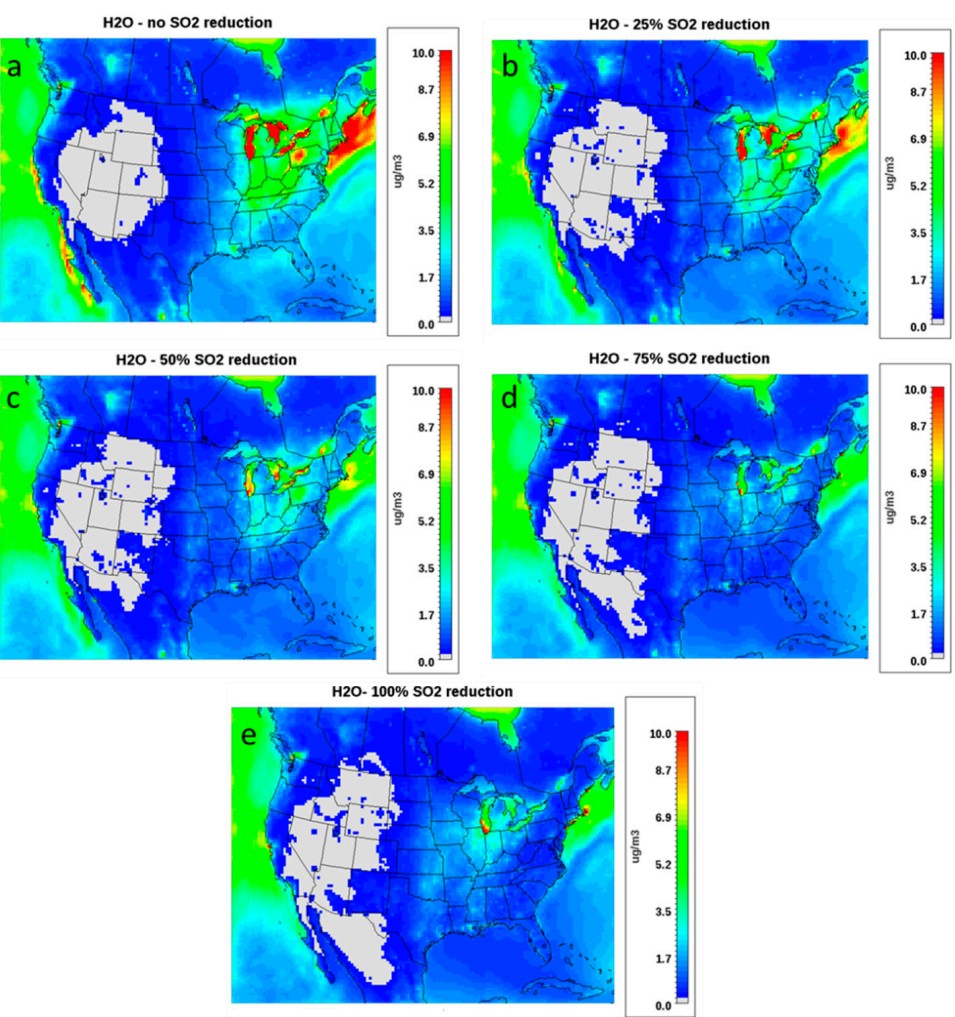

**Figure 8.** Simulated liquid water content (LWC) fields over the Continental United States (CONUS) for no reductions (**a**), 25% (**b**), 50% (**c**), 75% (**d**) and 100% (**e**) $SO_2$ emission reductions.

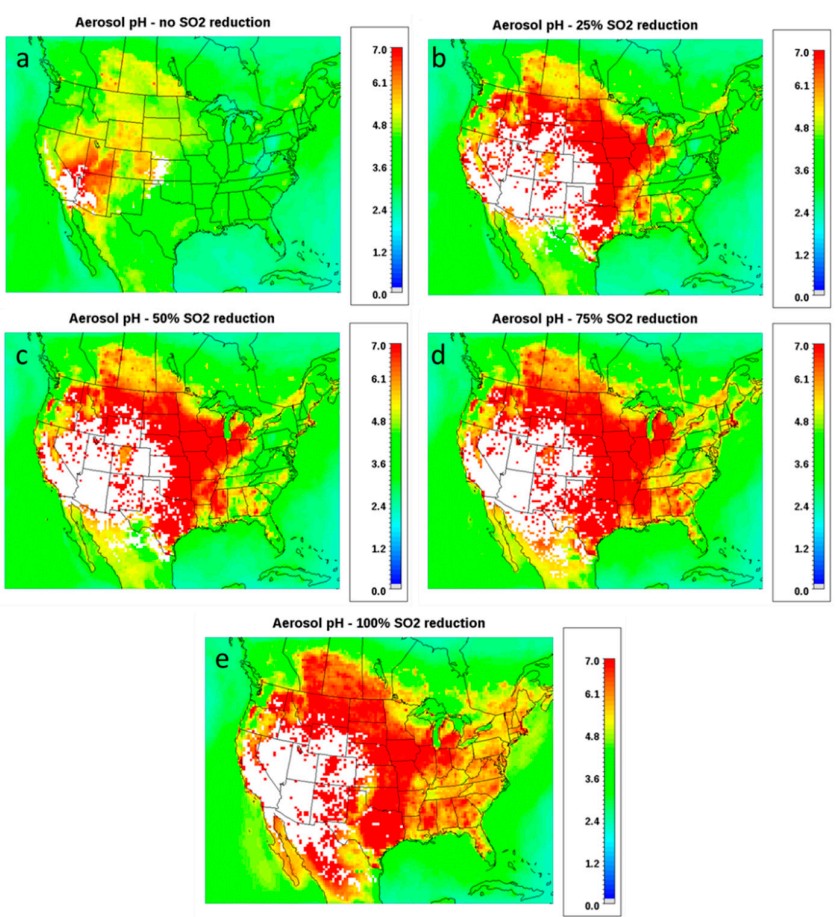

**Figure 9.** Simulated pH fields over the CONUS for no reductions (**a**), 25% (**b**), 50% (**c**), 75% (**d**) and 100% (**e**) SO$_2$ emission reductions. White areas have zero inorganic water and therefore pH is not predicted for them.

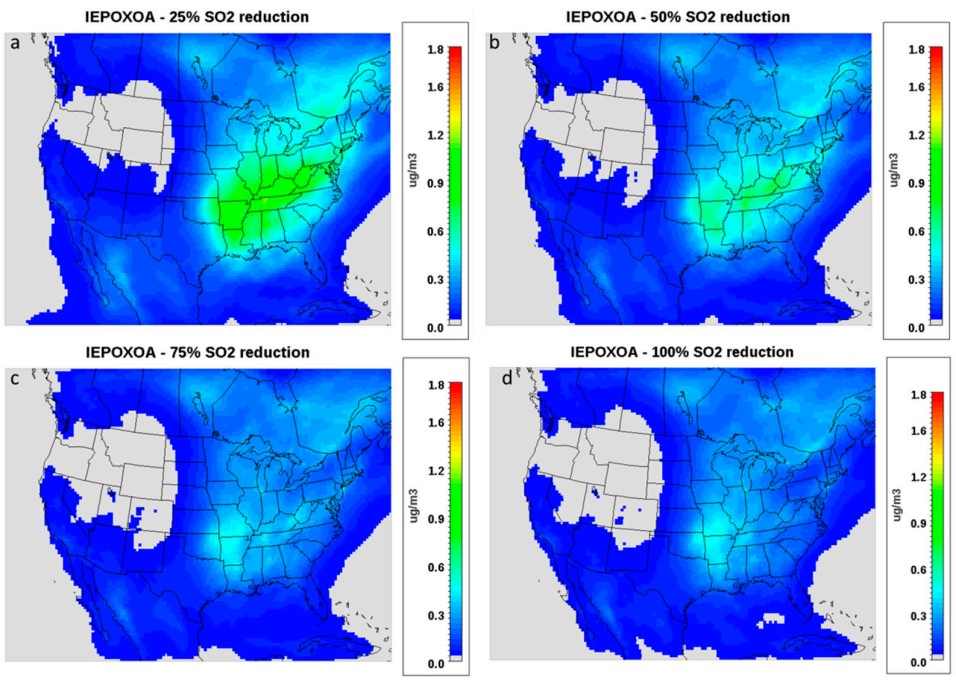

**Figure 10.** Simulated IEPOX OA fields over the CONUS for 25% (**a**), 50% (**b**), 75% (**c**) and 100% (**d**) SO$_2$ emission reductions.

The correlation between sulfate and IEPOX OA in the SOAS site, remains rather unaffected for all scenarios, and closely related to that one observed, apart from the case where $SO_2$ emissions are reduced by 100%, indicating that, even under the most stringent of reductions, sulfate concentration still remains the parameter that controls IEPOX OA formation (Figure 11). Specifically, the corresponding average increase of IEPOX OA ($\mu g\ m^{-3}$) to an increase of 1 $\mu g\ m^{-3}$ of sulfate is 0.67, 0.6 and 0.61 $\mu g\ m^{-3}$ for 25%, 50% and 75% reductions respectively (when compared to 0.5 $\mu g\ m^{-3}$ with no reductions). Under a 100% reduction scenario, the correlation becomes even more pronounced where an increase of 1 $\mu g\ m^{-3}$ of sulfate corresponds to a 1.27 $\mu g\ m^{-3}$ increase in IEPOX OA. These results indicate that even under less acidic conditions and the absence of sufficient LWC, the IEPOX pathway still remains active as long as there are even minute amounts of sulfate in the atmosphere, something that should be considered in policies aimed at reducing particulate matter ($PM_{2.5}$) levels.

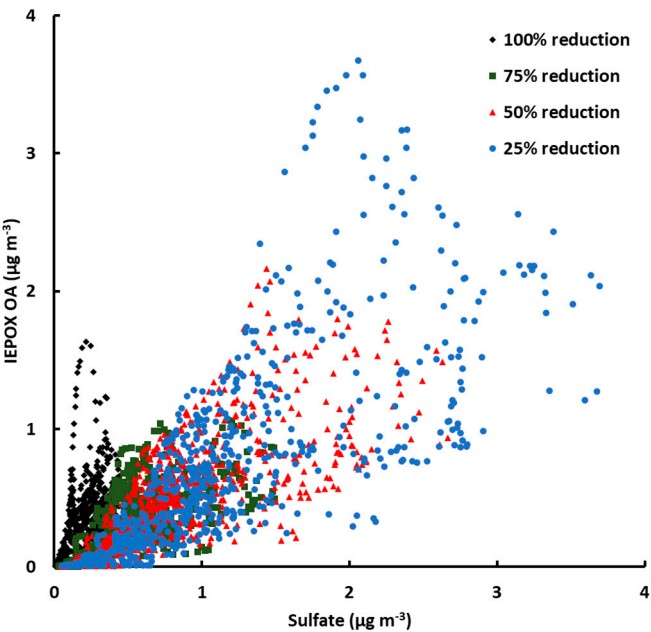

**Figure 11.** Correlation between sulfate and isoprene OA under 25% (blue), 50% (red), 75% (green) and 100% (black) $SO_2$ emission reductions.

## 4. Discussion

The study by Xu et al. 2015 underscored the importance of anthropogenic influence on isoprene OA formation and the potential of IEPOX OA to comprise a significant part of the total aerosol mass. Utilizing a CMAQ version with updated isoprene chemistry, we conducted simulations for the SOAS campaign period, and made model updates based on the extensive data set available from the campaign. We tested the ability of the model to reproduce the observed levels of IEPOX OA, as well as the relationship between IEPOX OA and sulfate.

The base configuration of CMAQ captured the dynamics of ozone, $NO_2$ and sulfate, although there was a positive bias in ozone and a negative bias in sulfate in the afternoon. Isoprene-derived SOA levels, as well as OA in general, were biased low. In addition, the current version of BEIS implemented in CMAQ, produced isoprene emissions which were dynamically inconsistent with the observed values. Model parameters and inputs that control the formation of isoprene-derived aerosols were varied to improve its representation of SOA formation, focusing on SOA resulting from IEPOX chemistry. In particular, isoprene emissions were adjusted to be more in accordance with the observed concentrations and fluxes and the treatment of IEPOX were made that led to IEPOX and other species were more in line with observations. The Henry's law constant for IEPOX was found to be of particular importance. A value of $1.9 \times 10^7$ M $atm^{-1}$ led to closest agreement between

modelled and observed concentrations of isoprene-derived SOA also captured the observed correlation of sulfate and isoprene OA.

The updates to the model also led to little correlation between isoprene OA and acidity ($H^+$) or particle water ($H_2O_{ptcl}$), consistent with recent studies [2,4]. The high aerosol water levels and low pH found in the SE US [15,16] is a possible explanation for the lack of correlation since at this regime they do not constitute limiting parameters. An increase of $H_2O_{ptcl}$ would be accompanied with dilution of $H^+$, negating the possible increase of IEPOX OA production. Another reason could be that SOAS took place during the summertime, meaning that nitrate partitioning that could potentially affect IEPOX OA production was not favored due to the higher temperatures. Higher temperatures could also prohibit the formation of viscous organic shells, thus making IEPOX OA formation less sensitive to variations in particle water and acidity, due to the removal of diffusion limitations in the transport of IEPOX from the gas to the aqueous phase.

NVCs were found to exert a potentially significant impact on IEPOX OA, through their propensity to increase aerosol pH and further promote the partitioning of inorganic nitrate. Given this and the known difficulties related to predictions of NVC levels across size in CMAQ, biases may be introduced in predictions of IEPOX OA (and many other acidity-sensitive processes; [17]), care should be taken to account for their effects when studying processes that rely on aerosol acidity.

As expected, $SO_2$ emission reductions limit LWC availability and lead to less acidic aerosol (when reduced more than 50%). However, IEPOX OA continues to respond linearly to sulfate even after drastic reductions, and tends to become more sensitive to perturbations in sulfate the greater the reductions are, indicating that even at a regime under which $H_2O_{ptcl}$ and $H^+$ are scarce, sulfate remains the main controlling factor for IEPOX OA formation.

Results underline the need for changes in the calculation of isoprene emissions in BEIS, and the more accurate representation of the physical processes that IEPOX undergoes during its lifetime. Laboratory experiments and field studies elucidating the formation of IEPOX triols, which are not included in the model, and comprise a significant portion of IEPOX OA, would also allow for a better prediction of the remaining isoprene OA fraction not predicted by CMAQ. In addition, the outstanding issue of elevated model NVC levels should be considered, given their ability to heavily influence pH predictions and subsequently IEPOX OA concentrations.

**Author Contributions:** Conceptualization, A.N., A.R., and P.V. Methodology, A.N., A.R., and P.V. Formal analysis, A.N., A.R., and P.V. Investigation, P.V. Writing—original draft preparation, P.V. Writing—review and editing, A.N., P.V., A.R. and Y.H. All authors have read and agreed to the published version of the manuscript.

**Funding:** This study was supported by the Environmental Protection Agency STAR Grant R835410, AN acknowledges support by the European Research Council, CoG-2016 project PyroTRACH (726165) funded by H2020-EU.1.1.—Excellent Science.

**Institutional Review Board Statement:** Not applicable.

**Informed Consent Statement:** Not applicable.

**Data Availability Statement: CMAQ** data will be made available at a later time due to their size (~1 Tb).

**Acknowledgments:** We gracefully acknowledge Havala Pye who provided the updated version of the CMAQ model.

**Conflicts of Interest:** The authors declare that they have no conflict of interest.

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
