# Peer review of "Determining the Role of Acidity, Fate and Formation of IEPOX-Derived SOA in CMAQ"

_atmosphere, doi:10.3390/atmos12060707_

Round 1

Reviewer 1 Report

Summary

This manuscript simulate the quantity of IEPOX-OA formation in the Southern East area of the United State by using the modified CMAQ  This paper discussed about potential uncertainties in predicting IEPOX-OA mass due to inaccuracy of sub-modules and model parameters such as PBL, isoprene emission, photochemical reactions of IEPOX, the concentrations of IEPOX, aerosol acidity due to neutralization by mineral dust cations, and the Henry’s constant of IEPOX.  The authors identified that the major issue in underestimation of IEPOX is related to the Henry’s constant of IEPOX.  This paper may be suitable for publication in Atmosphere but need major revision by improving the model assumption for the aerosol phase (single phase) and uncertainties in model parameters that were employed by the authors.  The some simulations were performed to show the sensitivity of IEPOX-OA to model parameters.  However, the simulated results need to be compared with field observation to convince the model performance. Please find the comments below.

Atmosphere review

1. Through the paper, the authors mainly discussed about the quantity of IEPOX-OA. The IEPOX-OA fraction of isoprene SOA or the IEPOX-OA fraction of ambient SOA needs to be discussed. There are other isoprene products that can attribute to isoprene SOA mass.  Thus, the discussion about the IEPOX-OA fraction of isoprene SOA can allow readers to understand the importance of IEPOX-OA.

2. Lines 69-72. The authors mentioned that aerosol pH is insensitive to SO2 reductions due to the buffering effect of semi-volatile ammonia that repartitions between the gas/particle phase in response to changes in sulfate. In order to increase pH, a large amount of ammonia. How much is a significantly low pH ?

3. Non-volatile cations (NVCs) was defined in Abstract but not defined in the main manuscript.

4. Lines 89-90. The authors mentioned that CMAQ exhibits a negative bias in modelled OA concentrations, especially SOA from isoprene oxidation. Provide the rationale about this discussion. It would be true the area where isoprene is dominant. Do you think that this fact be generalized all over the United State and apply to many other regions in other continents ?

If the model parameters were experimentally determined by using field data or laboratory studies, why did the model under-predicted isoprene SOA ?  Is it due to the bias in data itself or incorrect kinetic expression in gas and aerosol phase ?  

5 Lines 124-128. The authors mentioned that non-volatile cations have a large impact on pH and uncertainties in their emissions and size dependence can lead to biases in CMAQ. These data were removed from the simulations to avoid biasing the results.  In section 3.4, however, the authors discussed about the impact of NVC on IEPOX-OA.  Why do not the authors compare the simulation in the non-NVC periods with that in NVC periods ?  In Section 3.4, the simulation is a type of a sensitivity test.  There is no actual field data to prove the performance of simulation.

6 Lines 133-138. Did the authors used the same data that is collected in 2013 to improve the model ?  Are they the data that were also used in the simulation by using the preexisting CMAQ ?  The authors mentioned that IEPOX-OA was quantified by comparing predicted aerosol in their presence and absence.  It is unclear what the authors mean.  Do you mean the quantity of IEPOX-OA?  If the aerosol mass increases due to IEPOX-OA, the addition SOA mass can also increases mass due to gas-partitioning and subsequent of heterogeneous reactions of the organics.  How do the authors estimate IEPOX-OA mass ?

7. Lines 153-155. The authors mentioned that in all simulations including the base scenario, there is good agreement between the measurements and the simulation for atmospherically relevant compounds at the Centreville site. What do the overlapping areas mean in each species of Figure 2 ?  These areas are errors in field observation ? The deviations of predicted concentrations from observation are large for ozone and NOx.  It is hard to conclude that there was a good agreement between observation and simulation.  

  1. Section 3.2.2.

The authors proposes the potential uncertainties in both PBL and emission of isoprene. It seems that the emission of isoprene is more problematic.  The authors need to discuss how much PBL at daytime and nighttime was biased and how PBL bias and emission bias can be decoupled.  If the PBL at daytime is over-predicted, subsequently isoprene concentration can be under-predicted.  This should be clarified.

9 Lines 219-224.  The authors mentioned the dry deposition of IEPOX is problematic.  The authors also mentioned that the surrogate of Henry’s constant for IEPOX is inappropriate.  In general, the dry deposition includes the chemical and particle loss to soil.  It is confusing whether Henry’s constant is used for wet deposition or something else other than soil.  The authors need to define what is the subject for the dry deposition of IEPOX.  If the authors focus on aerosol aqueous phase, the authors may use word partitioning of organic species to aerosol aqueous phase.

10. The gas-aqueous partitioning onto the preexisting wet aerosol can be significantly impacted by inorganic salt compositions (ion species and aerosol acidity) and aerosol water content. In general, the Henry’s constant is equilibrium partitioning of organic species between gas phase and water. The partitioning of organic species to salted aqueous phase is considerably deviated from water (Henry’s constant).  How do the authors consider the actual partitioning of IEPOX to inorganic salted aqueous phase ?

11. Lines 235-244. Based on Figure 4, the gas-phase IEPOX concentrations were significantly overestimated in the model used by the authors (about 5 times).  How prediction in Figure 5 can be changed if the concentration of IEPOX is 5 times reduced.   

12. Figure 5. Is Figure 1 on page 7 mislabeled to Figure 5 ? Was Figure 1 wrongly placed onto page 7?

13 Lines 294-296.  How do the authors assume a single phase between inorganic salted aqueous phase and organic species ? If then, this assumption needs to be explained by a convincible observation or theory.  There are possibly less polar organic compounds from terpenes and anthropogenic sources.  This should be clarified.

14. The authors mentioned that aerosol acidity needs to be estimated based on only water. How can only water be separated in the homogenous single phase? The acidity of inorganic salted aerosol can also be influenced by hydrophilic organic compounds in the single phase. At least, inorganic species can be diluted by coexisting organic species.  This should be clarified.  In the paper, how do the authors estimate the aerosol acidity ?

15. Impact of aerosol acidity on IEPOX-OA.

At a given site, the variation of the sulfate to ammonia ratio may be small.  Because of the specific condition at SE, it is hard to conclude that there is a weak relationship between aerosol acidity and SOA mass.  Within a large scale of aerosol acidity due to very different inorganic composition (sulfate to ammonium ratio), the much clear relation can be appeared.

Were all the aerosols in Figure 5 wet (above the ERH or above the DRH)?  Is there a relationship between sulfate and IEPOX-OA due to the strength of photochemical reaction (sunlight intensity)?  Do both sulfate concentrations and the isoprene flux increase with increasing sunlight ?  During the daytime, isoprene flux increases but humidity drops.  This cycle has a diurnal pattern. Then, no clear humidity impact on SOA formation can be possible because sunlight intensity and temperature (humidity) can influence the intensity of a diurnal pattern.  

16. Section 3.4. The authors simulated the impact of dust on IEPOX-OA. Did the authors simulate IEPOX-OA simulation focusing on a specific field data to prove the impact of dust on SOA ?

Reviewer 2 Report

This manuscript utilized Community Multiscale Air Quality model with 2013 southern oxidant and aerosol study dataset and extensive mechanism of IEPOX-mediated SOA formation and impact of potential future emission controls on IEPOX-OA. Results show that Henry’s law coefficient for IEPOX plays the most significant roles in controlling aqueous isoprene OA products and that there is a strong correlation of isoprene OA with sulfate, and, in contrast, there is little correlation with acidity or liquid water content. The paper is well organized and written. Results are reasonable in agreement with literature and details are covered very well. The manuscript has a broad appeal and of great interest to atmospheric community. It merits publication as is.

Author Response

We gracefully thank the reviewer for his/her support and very kind review. We have addressed every point raised.

Reviewer 3 Report

This is a nice piece of modeling study showing the importance of sulfate in reducing IEPOX-OA, and consequently PM2.5 levels in the SE US. Overall, the manuscript is very well written and easy to follow. The conclusion is well supported by a series of results from well conducted model experiments. I ask the authors to address following two minor points prior to publication of this manuscript.

Line 61: As the ring opening reactions of IEPOX is catalyzed by H+, I’d expect higher aerosol acidity (i.e. lower aerosol pH) for IEPOX-OA formation.   

Line 303: Do you mean reduce pH (i.e. more acidic) due to higher hydrogen ion activity? It is a little unclear to me here.

Round 2

Reviewer 3 Report

The authors address all the issues raised by the reviewers adequately, and the manuscript meets the scientific quality standard set by the journal. I recommend this manuscript for publication as is.